# Thirty Years of Changes and the Current State of Swedish Animal Welfare Legislation

**DOI:** 10.3390/ani11102901

**Published:** 2021-10-06

**Authors:** Frida Lundmark Hedman, Charlotte Berg, Margareta Stéen

**Affiliations:** 1Department of Animal Environment and Health, Swedish University of Agricultural Sciences, P.O. Box 234, 532 23 Skara, Sweden; lotta.berg@slu.se; 2Department of Anatomy, Physiology and Biochemistry, Swedish University of Agricultural Sciences, P.O. Box 7011, 750 07 Uppsala, Sweden; margareta.steen@slu.se

**Keywords:** amendments, animal protection, farm animals, horses, regulations

## Abstract

**Simple Summary:**

Sweden is often cited as a leading country in animal welfare and related legislation, but some recent changes in the national legislation are seen as lowering the animal welfare requirements in order to improve the competitiveness of Swedish farmers. In this study, we analysed suggested changes to the Swedish welfare legislation between 1988 and 2019 relating to horses, cattle and pigs, including the written motivations, the written stakeholder responses and the actual changes to the final regulations. We used a sample of 77 legal requirements to assess in depth whether the animal welfare level was affected by these changes in the legislation. The results showed that the animal welfare requirements in Sweden for cattle, pigs and horses increased overall during the 30-year study period, but that a number of specific requirements had been relaxed to satisfy interests other than animal welfare. Thus, the new requirements should be evaluated more fully in order to determine whether they serve their purpose in practice.

**Abstract:**

Sweden is often seen as a leading country in animal welfare and legislation, but some recent amendments to the legislation are perceived as relaxing animal welfare requirements in order to improve the competitiveness of the relevant industry and of farmers. In this study, we analysed the suggested and actual changes in the Swedish national animal welfare regulations relating to horses, cattle and pigs between 1988 and 2019 and the consequences for the intended animal welfare level. The regulations and amendments, including the proposals, the written motivations, the stakeholders’ written responses to the proposed amendments and the final amendments, were scrutinised in detail. A sample of 77 requirements was then selected to assess whether and how the animal welfare level was affected by these legislative changes. The results indicated that the animal welfare protection level for cattle, pigs and horses increased overall during the 30-year period, but that a number of specific requirements had been relaxed to meet objectives other than animal welfare. It was more difficult to determine whether animal welfare improved in practice during the same period, due to the lack of systematic evaluations of the consequences of amending the regulations. Future evaluations are needed to evaluate the outcome of new legislative requirements and to monitor whether they serve their purpose in practice.

## 1. Introduction

Sweden is often cited and categorised as a leading country with respect to animal welfare in general and animal welfare legislation in particular [1]. Sweden has had requirements regarding allowing animals to perform their natural behaviours since the Animal Welfare Act of 1988 (SFS 1988:534), including mandating summer pasture for dairy cows and banning sow crates and battery cages for laying hens. The most recent Animal Welfare Act in Sweden (SFS 2018:1192), introduced in April 2019, pushes the animal welfare position further. It states not only that animals should be allowed to express their natural behaviours and be protected against unnecessary suffering but also that good animal welfare should be ensured, animal well-being should be promoted and all animals should be shown respect.

Animal welfare legislation in Sweden is enacted at three levels. The Swedish Parliament is responsible for the Animal Welfare Act, where the overall aims and frames concerning animal welfare are set. The government is responsible for the Animal Welfare Ordinance, which is also written in quite general terms. The Swedish Board of Agriculture (SBA) is the current central competent authority (CCA) responsible for the national regulations concerning animal welfare. These regulations are much more detailed than the Animal Welfare Act and Animal Welfare Ordinance and contain requirements concerning the housing and management of different animal species; not only farm animals but also companion animals, animals in sports and zoo animals. The CCA in Sweden is an autonomous expert body and the government cannot interfere with its decision-making or regulations. The SBA has been the CCA on animal welfare since 1991, except during the period 2003–2007, when the Swedish Animal Welfare Agency (SAWA) had this responsibility.

The national regulations have been modified by the CCA several times since 1988. Some of the more recent amendments have been criticised for relaxing the animal welfare requirements too much in favour of producers or the relevant industry. For example, since 2017, piglets may be weaned earlier than at 28 days of age (which was not permitted previously) and changes have been made in the pasture requirements for dairy cows, in order to make the legislation more flexible. Stakeholders representing animal welfare organisations, consumer organisations, politicians and researchers have expressed concerns that the animal welfare level in Sweden is decreasing due to the amendments made to the national regulations [2,3,4,5,6,7,8]. This study examined whether such concerns are justified, or whether the regulations have improved animal welfare during recent decades. This was achieved by analysing how the Swedish animal welfare regulations changed between 1988 and 2019 and the consequences for the intended animal welfare level.

## 2. Materials and Methods

The animal welfare regulations issued by the CCA between 1988 and 2019 regarding cattle, pigs and horses and other proposed amendments to these regulations (28 in total) were scrutinised in detail (Table 1). The documents analysed included proposals for amendments, the CCA’s written consequence analyses and justifications for the proposals, stakeholders’ written responses to the suggested amendments and the final amendments. The regulations included covered animal management and housing at the farm level, i.e., not transport or slaughter. From these regulations, we selected for detailed analysis a sample of 77 requirements meeting one or more of the following criteria: the suggested requirement was completely new; it involved major changes for animal husbandry and animal welfare; it generated many comments from stakeholders; the requirement was changed considerably between the proposal and the final amendment. Each amendment was then assessed in terms of whether and how it actually changed the animal welfare level for the species concerned. The developments or modifications of the legislation were judged as “improving animal welfare” if they included new or extended requirements in line with generally acknowledged and scientifically based actions or inputs related to improved animal welfare, such as increased space allowance, improved possibilities for social contact with conspecifics or other important behavioural needs, stricter requirements for access to feed and water, the introduction of stricter animal-based indicators and so on. Similarly, the developments or modifications were judged as “impairing animal welfare” if any of the requirements listed above were relaxed or removed from the legislation.

The inputs and comments on the proposed modifications and amendments to the legislation provided to the central competent authority for animal welfare by stakeholders are described in detail in Lundmark Hedman et al., 2021 [9].

## 3. Results

### 3.1. Amendments Relating to Horses, Cattle and Pigs

The CCA has updated the animal welfare regulations for cattle, pigs and horses multiple times since 1988, with 20 updates for cattle, 16 for pigs and 11 for horses. Prior to 1989, there were no binding regulations concerning these species, and animal owners only had to consider the CCA’s general advice. Cattle, pigs and horses were covered by the same regulations until 2007, when the requirements for horses were compiled in a specific set of equine regulations. Since 2017, pigs and cattle have had separate sets of regulations.

Horses: Animal welfare regulations for horses were very scarce in 1989, and horses were excluded from most of the existing requirements concerning animal housing and management until 1993. The requirements were extended in 1993 to cover horses, stating, e.g., that horses must be kept clean, given daily attention and supervision, have well-trimmed hooves, have access to nutritious feed and have a dry and clean stable with a satisfactory indoor climate. However, only horses bred and kept for “competition purposes” were covered by the regulations at that time, while for other horses, the requirements were intended only as general advice. In 2003, the regulations were extended to all horses, irrespective of the purpose for which the horses were kept. The most comprehensive amendments concerning horses were made in 2007, when the CCA (SAWA) introduced additional requirements regarding enabling horses’ need for social contact, providing daily exercise in paddocks or similar areas, limiting the time for which horses could be tied up in stalls and banning the building of new stables with tie stalls. In 2018, the CCA (SBA) revised the horse regulations more structurally than contextually, in order to make them more goal-oriented and flexible and less detailed. Since 1989, the number of animal welfare requirements concerning horses has increased from 4 to 50.

Cattle: Several of the requirements for cattle in the regulations of 1989 are still present, more or less unmodified, in the current regulations. However, new requirements have been added and amendments made relating to, e.g., the mandatory summer pasture period (1995, 1997 and 2016), the dimensions of stalls and houses and bans on constructing new houses with tie stalls (2007). In 1997, the national regulations were amended in order to implement the EU Directive on the keeping of calves (91/629/EEG). Since 1989, the number of animal welfare requirements concerning cattle kept on farms has increased from 45 to 79.

Pigs: There have been greater amendments to the regulations for pigs than to those for cattle during the past 30 years. The most comprehensive amendments were made in 2006, with new requirements requiring the provision of straw for all pigs and nest-building material for sows before farrowing. In 2010, the SBA implemented an amendment requiring all pigs to be group-housed, with some exceptions for boars and farrowing sows. In 2017, the regulations were relaxed to allow piglets to be weaned prior to 28 days of age under certain conditions. Since 1989, the number of animal welfare requirements concerning the keeping of pigs has increased from 38 to 59.

### 3.2. Animal Welfare Level in Suggested Amendments to the Regulations

For the 77 legal requirements scrutinised in this study, 47% of the suggested amendments involved a higher animal welfare level, 8% a lower animal welfare level, 8% an unchanged level of animal welfare and 17% both lower and higher animal welfare levels in the same requirement. For 19% of the suggested amendments, it was not possible to determine clearly whether they involved any increase or decrease in animal welfare.

Examples of an improved animal welfare level were: suggestions requiring hoof trimming in pigs (SJVFS 1993:129); free access to water for pigs (SJVFS 2003:6); daily exercise in paddocks for horses (SJVFS 2003:6 and DFS 2007:6); a ban on the use of barbed wire as fencing in horse paddocks (DFS 2005:11); straw for manipulation, occupation and comfort behaviour for pigs (DFS 2006:4); nest-building material for sows before farrowing (DFS 2006:4); a ban on building new stables with tie stalls for horses, including a restriction on keeping horses tied up for more than 16 h per day (DFS 2007:6); a ban on building new houses with tie stalls for cattle, including, from 1 June 2017, a total ban on having male cattle tied up (DFS 2007:5); restrictions on the use of slatted floors for cattle and a requirement for slatted floors to be covered with rubber or other resilient material (DFS 2007:5); free access to water for horses at temperatures above 0 °C (DFS 2007:6); fulfilment of a horse’s social needs (DFS 2007:6), normally by keeping it together with another horse (SJVFS 2018:49); a ban on trimming or removal of horses’ whiskers (DFS 2007:5 and SJVFS 2018:49); a requirement that cattle should be kept together with other cattle (SJVFS 2014:31).

Amendments suggesting a lower animal welfare level were: allowing barbed wire fences to be combined with electric fences for cattle (DFS 2005:9); further exceptions to cattle being out on summer pasture (SJVFS 2003:6, SJVFS 2012:13 and SJVFS 2016:13); allowing the weaning of piglets before 28 days of age (SJVFS 2017:25); lowering of the space requirements, either through smaller pen sizes or by allowing a higher stocking density (although increased space requirements were suggested in some cases).

In the above cases, the suggested changes were sometimes difficult to categorise as a clear improvement or reduction in animal welfare. Another category of amendments that were difficult to assess and categorise with respect to whether they increased or decreased animal welfare were the requirements that had been rewritten from being quite precise and detailed to being more goal-oriented and flexible. For example, the water requirement for horses previously gave the general advice that automatic water cups should have a flow of at least 6 litres per minute, and this was changed to stating that automatic water cups must have “a large enough” flow (SJVFS 2018:49). In 2017, the CCA also suggested that the requirement stating the precise maximum levels of different air pollutants in horse stables should be removed and replaced with a requirement stating that horses must not be exposed to air pollution at a level that can “adversely affect their health”. Over the years, there have also been suggestions regarding replacing the specified space and dimension requirements with instructions on what to achieve in more general terms, rather than in terms of minimum lengths/areas.

### 3.3. Animal Welfare Level in the Final Amendments

After the referral round, where different groups of stakeholders were given the opportunity to express their views on the suggested amendments, the requirements were most commonly not changed by the CCA, i.e., 66% of the final amendments had the same animal welfare level as stated in the suggested amendments. However, the animal welfare level increased in 12% of the final amendments compared with the proposals, while it decreased in 9%. For 12% of the final amendments, it was not possible to assess whether the change between the suggested and final wording meant an actual difference for the animal welfare level.

The final required level was decreased, e.g., for horses in 2003, when the suggested requirement for daily exercise in paddocks was not implemented (SJVFS 2003:6). This requirement was instead implemented in 2007, but several exceptions governing when horses could be kept indoors were added to the final regulations (DFS 2007:6), and hence the animal welfare level was lower in the final amendment than in the proposal. A similar outcome was observed concerning the suggested ban on the removal of horses’ whiskers, which was not implemented in 2007 but instead downgraded to general advice (DFS 2007:5). However, the removal of horses’ whiskers was finally banned in 2018 (SJVFS 2018:49). The suggested requirement for free access to water for horses at temperatures above 0 °C was not implemented at all, and instead, the old requirement of providing water at least twice a day was retained, together with new general advice that horses should have free access to water (DFS 2007:6).

An increase in the animal welfare level between the suggested and the final amendment was observed in the following cases: when the CCA removed the suggestion that County Administrative Boards (i.e., regional authorities) should be able to grant exemptions to the requirement for shelters for animals kept permanently outdoors during winter (SJVFS 1993:129); when the minimum proportion of solid floor increased from 50% to 80% for tied cattle (SJVFS 2003:6); when all types of horses were covered by the regulations, not only horses kept for racing and competition purposes (SJVFS 2003:6); when the CCA removed the suggestion that being “outdoors” could replace some of the time cattle should spent on pasture (SJVFS 2016:13). The suggestion in 2017 to replace the specific maximum levels of different air pollutants in horse stables with a more goal-oriented requirement was not implemented, so the specified maximum levels still remain (SJVFS 2018:49).

## 4. Discussion

### 4.1. Improved Animal Welfare Level in National Regulations

This analysis shows that the animal welfare level required in the national regulations for horses, cattle and pigs in Sweden has increased in general during the past 30 years, with the changes being most comprehensive for horses and their owners. However, there are some challenges when comparing animal welfare levels, so it is important to consider the different aspects that must be kept in mind when comparing animal welfare levels in the legislation.

### 4.2. The Legislation Does Not Strive for Best Possible Animal Welfare

The regulations set out the minimum level of animal welfare, not the best possible level, so a change in the regulations may mean that one minimum requirement is replaced with another minimum requirement. For example, the requirements concerning slatted floors for cattle have become more detailed and specific, and slatted floors must now be covered with rubber or other resilient material. This is better for cattle than a slatted floor made from concrete [10,11] but is still not optimal from an animal welfare point of view [12,13].

### 4.3. The Level of Change Will Depend on the Definition of Animal Welfare Used

Whether a change is seen as entailing a higher, lower or maintained level of animal welfare can depend on how the concept of “*animal welfare*” is defined. Different individuals and groups may place different emphases on various aspects of animal welfare, e.g., biological functioning, leading natural lives or feelings, leading to diverse conclusions [14]. In the present analysis, it was sometimes difficult to assess whether an amendment would lead to improved or deteriorated animal welfare. For example, the dimensions of pens were decreased and slatted floors were permitted in order to improve pen and animal hygiene, but this also restricts the animals’ ability to move freely and behave naturally. It is therefore difficult to state whether these alterations lead to an overall improvement in the welfare of the animals or whether the negative aspects will outweigh the improvements, with the final outcome partly depending on the level of importance assigned to the different aspects.

### 4.4. An Improved Protection Level Does Not Necessarily Mean Improved Animal Welfare

Animal welfare legislation is designed for preventative purposes and consists mainly of requirements relating to interior animal house design and animal management, i.e., resource- and management-based requirements, in order to detect and limit welfare risks [15,16]. While it can be assumed that many of the improvements in the regulations lead to better animal welfare in practice at the farm level, the amendments would need to be evaluated more systematically and in greater detail in order to verify this correlation. According to statistics from the official animal welfare control authorities in Sweden, cattle not being “clean enough” is a commonly recurring form of non-compliance and has been so for several years [17,18]. Hence, animal welfare at the farm level has not improved with regard to this welfare indicator, despite amendments to the regulations. In order to evaluate the effect of a specific amendment on actual animal welfare level on farms, more animal-based indicators would need to be used in monitoring systems.

### 4.5. Animal Welfare Level Is Often Related to What Others Have Done

The animal welfare level in Sweden is often compared with that at the European Union (EU) level or that in other countries. Hence, animal welfare is often measured in relation to what others do and not necessarily in relation to the best possible level of animal welfare. Comparisons with others are not necessarily negative, since they may encourage and motivate those with poorer animal welfare to make improvements [19,20]. However, they can also be used to justify reductions in animal welfare, e.g., when the SBA states that the younger acceptable weaning age for piglets in Sweden is still better than in the rest of the EU [21], even though science cannot support such a relaxation of the regulation from the piglet welfare perspective [22].

### 4.6. Awaiting Stakeholder Acceptance

Previous work has shown that the process of increasing animal welfare levels through stricter regulations can be impeded if the relevant industry is opposed to the higher requirements [9]. This is especially evident with regard to the animal welfare regulations for horses. For example, the CCA suggested in 2003 that horses must be allowed to go outdoors in paddocks or on pasture on a daily basis, but the horse industry was very critical of the idea, and the suggested amendment did not come into force. In 2007, the CCA tried again and the horse industry was still sceptical, but this time the CCA pushed through the new regulations, although in the final amendment, several exceptions to outdoor access were added. This is one clear example of the CCA balancing the interests of animals against those of the industry and waiting for greater acceptance from stakeholders before increasing the animal welfare level in the regulations. It is also an example of the CCA, despite criticism, slowly and at least partly forcing an industry to change how animals are kept and managed.

### 4.7. Moving towards More Goal-Oriented and Flexible Regulations

In most European countries, the central competent authorities are organised directly under the control of the ministries, or are incorporated into the ministries and are under direct ministerial rule. The Swedish approach, with independent authorities, leaves animal welfare legislation less exposed to rapid political changes and the personal preferences of ministers or their political secretaries, making it easier for the authorities to maintain a steady direction in their policy work. However, this also grants a reasonable amount of executive power to the authorities, which may be slow in responding to public demands for changes in the animal welfare legislation as they will not be held accountable at the next election in the same way as politicians. Parliament, the government and the relevant minister are of course still able to influence the legislative process via changes in the national Animal Welfare Act or Animal Welfare Ordinance or via various types of directions and decrees to the authorities [9]. Decisions made by the competent authorities, at the central or regional level, can be challenged by a standard court procedure [23]. In recent years, the government directions to the CCA have focused on creating regulations that are more goal-oriented and flexible [9]. How this rephrasing will affect the actual animal welfare levels required and observed in practice is not clear and has not been evaluated, but the purpose (according to the CCA) is to maintain the same level of animal welfare. However, some argue that goal-oriented, flexible regulations leave more room for misinterpretations, due to vague wording such as “enough”, “normally” and “if possible”, as the legislation does not strive for the best possible animal welfare [24,25,26,27,28,29], and therefore diverging assessments and interpretations may be made. The horse industry and several other stakeholders seem to be aware of this risk and have expressed fear of the consequences, but the industry in general is very positive towards goal-oriented, flexible regulations [9].

### 4.8. Mind the Gaps

Animal welfare regulations exist to protect animals, but amendments to the regulations are not based solely on new scientific findings relating to animal welfare per se. On the contrary, several other aspects, such as politics, economics, geographically conditioned land and climate constraints and society’s views, also affect the legislation [30,31,32]. According to its government mandate, in addition to striving for good animal welfare, the SBA must also improve the competitiveness of Swedish farmers and enable an increase in Swedish food production [33]. Such directions will sometimes result in relaxation of the animal welfare regulations. However, there is a risk of the welfare levels presented in the Animal Welfare Act being perceived as more far-reaching than the detailed minimum requirements stated in the CCA’s regulations, i.e., in the second-level legislation [34]. Similar gaps between ambitious legislative animal welfare goals and their practical implementation have been identified regarding EU legislation [35] and Australian legislation [36]. Previous studies have also shown that animal welfare inspectors do not focus on the overall intentions and requirements in the Animal Welfare Act, e.g., the requirement on allowing animals to perform natural behaviours, but only on the detailed requirements in the second-level legislation [28,37]. Hence, if the CCA regulations do not replicate the animal welfare requirements stated in the Act, the intended level of animal welfare will not be realised. Furthermore, the new Swedish Animal Welfare Act (2018:1192) has an even more far-reaching purpose than the 1988 Act, because it seeks to ensure good animal welfare and promote animal well-being and respect for animals, i.e., Sweden has moved from a “welfarist view” to an “intrinsic value view” [38]. Some other European countries, e.g., Switzerland and Norway, have also adopted an intrinsic value view, using wordings such as “respect for dignity” and “respect for animals”, with a clearer aim of protecting animals for their own sake [38]. In practice, however, the intrinsic value view still tends to be similar to the welfarist view [24,38,39], and the consequences of the new Swedish Animal Welfare Act do not seem to be very extensive [40].

## 5. Conclusions

This study shows that the animal welfare protection level in legislation covering Swedish cattle, pigs and horses has increased in recent decades, although some requirements have been relaxed to meet non-animal-welfare interests. However, it is more difficult to say whether animal welfare has improved in practice during the same period, because there have been no systematic evaluations of the consequences of amendments to the legislation. Thus, evaluations are needed to determine the outcomes of any new requirements in practice and to assess whether they serve their purpose.

This study does not point to an ideal way of writing or amending legislation, nor does it identify an ideal level of animal protection to be specified in legislation. The results can nevertheless serve as a useful tool for public or private policymakers to improve transparency in the commentary process and reveal the delicate balance between different interests in this process.

## Figures and Tables

**Table 1 animals-11-02901-t001:** Swedish animal welfare regulations on the housing and management of horses, cattle and pigs analysed in this study. ‘LFS’ represents regulations made by the National Board of Agriculture, ‘SJVFS’ represents regulations made by the Swedish Board of Agriculture and ‘DFS’ represents regulations made by the Swedish Animal Welfare Agency.

Regulation	Species Covered by the Amendment
LFS 1989:20	Horses, cattle and pigs
SJVFS 1993:129	Horses, cattle and pigs
SJVFS 1994:2	Horses, cattle and pigs
SJVFS 1995:93	Cattle
SJVFS 1997:30	Cattle
SJVFS 1997:124	Cattle
SJVFS 1998:25	Cattle
SJVFS 2000:107	Horses, cattle and pigs
SJVFS 2003:3	Horses, cattle and pigs
SJVFS 2003:6	Horses, cattle and pigs
SJVFS 2003:41	Horses, cattle and pigs
DFS 2004:17	Horses, cattle and pigs
DFS 2005:9	Horses, cattle and pigs
DFS 2005:11	Horses, cattle and pigs
DFS 2006:4	Pigs
DFS 2007:5	Cattle and pigs
DFS 2007:6	Horses
SJVFS 2010:15	Cattle and pigs
SJVFS 2010:77	Pigs
SJVFS 2012:13	Cattle
SJVFS 2014:31	Cattle and pigs
SJVFS 2016:13	Cattle
SJVFS 2017:24	Cattle
SJVFS 2017:25	Pigs
SJVFS 2018:49	Horses
SJVFS 2019:17	Horses
SJVFS 2019:18	Cattle
SJVFS 2019:20	Pigs

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
