# Peer review of "Thirty Years of Changes and the Current State of Swedish Animal Welfare Legislation"

_animals, 2021, doi:10.3390/ani11102901_

Round 1
Reviewer 1 Report
This is an interesting case study of changes in animal welfare legislation in Sweden. Whilst the results are limited to Sweden this study is a novel approach to describing animal welfare policy which should be welcomed and could stimulate other such studies.
It is interesting to read about independent authority making difficult decisions balancing industry & animal welfare interests. It would be useful to compare this approach to other countries (& EU) where the detail regulations are under political control. i.e. what are the pros & cons of the 2 approaches ? What mechanisms are in place to challenge decisions etc ?
The results are presented as a text commentary - It would have been useful for a table to describe the animal welfare impact of the potential & actual changes - this would verify the text summary
Author Response
We have included some reasoning regarding this approach under section 4.7. We have also added two more references regarding implementation gaps in the discussion.
We considered summarizing the results in a table as suggested by Reviewer 1, but found this very complicated and unfortunately not helpful to the reader. Hence, we have chosen not to include such a table in this paper.
Reviewer 2 Report
The inputs and evaluations from the stakeholders are in a black box for the readers - both the methods, validity and results. I would suggest that you describe and assess this topic.
Author Response
The inputs and evaluations from the stakeholders have already been presented in another article, so we have tried to make this clearer in the material and method section.
Reviewer 3 Report
The idea of evaluating a country's performance regarding animal welfare is a good one. However, your study lacks a reliable method by which this done that would provide a little objectivity. It is not supported by, for example, a philosophical position by which this can be done. There is not scientific measure to whether the various new law can be judge according to an objective measure of increasing animal welfare. You need to explore a way in which some sort of scientific process could be applied to new laws as they emerged. You can't just rely on the simple proposition that they didn't do enough. E.g. did they advance in relation to new notions of 'sentience'?
In addition - check your English - lots of simple typos.
Author Response
The approach of this paper is not to present a solution to a given problem, rather to analyse the current state of practice and describe a processes. You (reviewer 3) are suggesting a completely different paper, which may certainly be interesting but which would still have to be based on this present paper. Hence, we are unable to take these comments into account in this particular paper, but will bear them in mind when continuing this line of research. This paper is analytical and descriptive, not pretending to have solutions for the problems just identified. However, we have added a few sentences on this to our conclusion chapter.
The paper has been sent to a professional English proofreader.
Round 2
Reviewer 2 Report
I have now been reading the revised version and the comments. My conclusion is, that it has been sufficiently improved.Author Response
We have added an explanation in the method section (line 87-94) concerning the judgement of the development of the animal welfare level of the legislation.
Reviewer 3 Report
I am still concerned that there is not enough on methods - by what are the developments being judged? If a section on this is included then publication would be warranted.
Author Response
We have added an explanation in the method section (line 87-94) concerning the judgement of the development of the animal welfare level of the legislation.